# Anxiety, Depression, Self-Esteem, Internet Addiction and Predictors of Cyberbullying and Cybervictimization among Female Nursing University Students: A Cross Sectional Study

**DOI:** 10.3390/ijerph20054293

**Published:** 2023-02-28

**Authors:** Zainab Fatehi Albikawi

**Affiliations:** Community and Psychiatric/Mental Health Nursing Department, Nursing College, King Khalid University, Khamis Mushait 39746, Saudi Arabia; zalbikawi@kku.edu.sa

**Keywords:** cyberbullying, nursing students, cybervictimization, mental health, internet addiction, self-esteem

## Abstract

Background: Cyberbullying and cybervictimization, which have been linked to the growth of the Internet and issues with mental health, can have serious psychological and academic consequences for young individuals, yet they have received relatively little scientific attention at universities. These phenomena have become an alarming social issue due to their rising rate and devastating physical and psychological effects on undergraduate university students. Aim: to estimate the prevalence of depression, low self-esteem, cybervictimization, anxiety, cyberbullying, and Internet addiction among Saudi female nursing university students and to identify the factors that predict cybervictimization and cyberbullying. Methods: Convenience sampling was used to select 179 female nursing university students with an average age of 20.80 ± 1.62 years for the purpose of conducting a descriptive cross-sectional study. Results: The percentage of students who reported having low self-esteem was 19.55%, depression (30.17%), Internet addiction (49.16%), anxiety (34.64%), cyberbullying (20.67%), and cybervictimization (17.32%). There was an inverse relationship between students’ self-esteem and their risk of engaging in cyberbullying (AOR = 0.782, 95% CI: 0.830–0.950, p = 0.002) or becoming cybervictims (AOR = 0.840, 95% CI: 0.810–0.920, *p* < 0.001). Further, Internet addiction predicted both cyberbullying (AOR = 1.028, 95% CI: 1.012–1.049, *p* = 0.003) and cybervictimization (AOR = 1.027, 95% CI: 1.010–1.042, *p* < 0.001). The likelihood of experiencing anxiety was linked to cyberbullying (AOR = 1.047, 95% CI: 1.031–1.139, *p* < 0.001) and cybervictimization (AOR = 1.042, 95% CI: 1.030–1.066, *p* < 0.001). Conclusion: Importantly, the findings suggest that programs designed to help university students avoid participating in cyberbullying activities or becoming cybervictims should take into account the influence of Internet addiction, mental health issues, and self-esteem.

## 1. Introduction

The widespread adoption of information and communication technology, particularly among younger generations, has fundamentally altered the manner in which our society consumes information, and, as a consequence, the ways in which we communicate and engage with one another. There is no question that this has some positive effects; nevertheless, there are also certain issues that have recently surfaced due to the improper application of these innovative technologies [1]. Over 59% of the world’s population makes use of the Internet, making it one of the most widely utilized and significant communication channels. Furthermore, this rate can get as high as 95% in developed countries [2]. According to recent research, 99.2% of college students use the Internet, with 24.8% exhibiting online addiction [3]. On the other hand, a recent survey conducted among college students found that 6.8% of them engage in cyberbullying, while 29.2% of them are victims of cyberbullying, and 28.2% of them were not active in either [4]. Moreover, students at universities encounter not only academic but also additional psychological problems [5].

### 1.1. Cyberbullying and Cybervictimization

According to previous studies, there is currently no universal definition of cyberbullying across the world. Cyberbullying is the repetitive use of technology such as e-mail, mobile phone messaging, social networks, websites, chat rooms, and instant messaging for the purpose of causing harm to another person [6]. Cyberbullying, in its broadest sense, refers to the intentional harassment or harming of another person through the use of electronic means of communication. This can be done in a variety of ways, such as through the use of text or online messages, social media comments or posts, online image postings that are meant to be demeaning, or by the use of electronic threats or intimidation [7,8]. Regrettably, cyberbullying has become a common practice among adolescents. Cyberbullying is distinct from more traditional forms of harassment because it has the potential to reach an unlimited amount of individuals at once, and across vast geographic and temporal distances. It is also a more permanent form of storage that stores text and photos without any oversight. Because cyberbullying perpetrators rarely, if ever, encounter their victims face-to-face, they often fail to grasp the full responsibility of their actions and are less likely to feel accountable for them [9].

Cyberbullying can cause adolescents to suffer from heartache, shame, humiliation, and marginalization. Recently, this has come to the attention of many researchers. Additionally, some adolescents are motivated by revenge [10]. University-level cyberbullying can be considered as a transitional stage in the continuum of this behavior from childhood and adolescence to adulthood [11]. As more developing nations have access to and use the Internet, this trend is growing [12]. While most prior research on cyberbullying has been conducted with younger teenagers, more recently, researchers have started looking at the topic with older adolescents as well, specifically, university students. Given that university students are among the most regular users of digital technology, it is logical that this new focus of research be directed in their direction. Other than psychological harm, cyberbullying might have much worse effects. Even though life and the Internet are worlds apart, some of the more severe instances of cyberbullying can escalate into actual violence when victims believe there are no other ways to resolve their disputes [13,14]. According to studies, cyberbullying and actual bullying are positively connected, which implies that those who bully others online are also more inclined to bully others in person, and vice versa [15,16]. To put it another way, cyberbullying is the act of bullying others, whereas cybervictimization is the act of being bullied online by others.

Researchers have, up until this point, incorporated the experience of victimization into their conceptualizations of cyberbullying incidents [17,18]. The term “cyberbullying victimization” refers to the act of one or more people receiving communications that constitute cyberbullying [19]. “You’re unattractive,” “you’re foolish,” “go harm yourself,” and similar insults are all examples of the types of unpleasant comments that can be sent via electronic communication or social media and constitute cyberbullying. The concept of cybervictimization is conceptually distinct from the concept of traditional victimization in that cyberbullying victimization necessitates the use of communication technology and/or the Internet to communicate aggressive messages [20]. An anonymous perpetrator of cyberbullying, who may or may not be known to the victim and who is not physically present in the victim’s environment, is another form of victimization that can result from this type of behavior [20]. In a recent study, cyberbullying and victimization are found to be common problems among college students, with 57% and 68% of students reporting these problems, respectively [2].

### 1.2. Female University Students and the Study Variables

In the field of nursing, women tend to be more common than men. About 75% of Saudi Arabia’s nursing staff are women [21]. Among all Internet users, a larger percentage of women (71% vs. 62%) utilize social networking sites than men [22]. Moreover, the majority of female undergraduates (57%) reported being cybervictimized, while a minority of males (43%) reported the same [23]. Additionaly, Choi and Lee [24] discovered in a self-report study that being a female college student is a predictor of being a cybervictim. Nearly half of all female college students (44%) had experienced cyberbullying, according to a study conducted among female college students. Moreover, over three-quarters of female college students (75%), including 36.3% in the severe to extremely severe range, experienced depression; more than half of male college students (53.8%), including 16.9% in the severe to extremely severe range, did so. However, while only 66.2% of male college students experience anxiety, this issue is reported by 80.6% of female college students [25].

### 1.3. Cyberbullying and Cybervictimization and Study Variables

Prior research [15,26,27] demonstrated that the incidence of cybervictimization ranged from 2.7% to 84.9%, whereas the incidence of cyberbullying ranged from 2.0% to 43.7%. Added to that, earlier research [28,29] has demonstrated that gender, self-esteem, anxiety, and Internet addiction may be somewhat linked to Internet bullying. Psychological impacts studies on cyberbullying have shown that its victims frequently struggle with depressive symptoms, anxiety, low self-esteem, antisocial actions, and even suicidal thoughts and intentions [30,31,32,33]. Furthermore, nursing students typically report suffering from mental health issues such as depression and anxiety [34].

Depression is a growing problem at community colleges, and it is a major contributor to disability. Multiple international studies found that depression among college students ranged from 15.6% to as high as 33% [35,36,37]. However, cyberbullying has consequences beyond the victims themselves. Even bullies can suffer negative effects from engaging in cyberbullying. Perpetrators have been shown to suffer from elevated stress, poor academic performance, increased vulnerability to depression, and alcohol usage [27,38]. 

One of the issues that can arise as a result of bullying is anxiety, which can be thought of as an unpleasant affective state. This issue can impact both the bully and the victim [39]. Both traditional and cybervictimization were found to have a positive link with anxiety [17]. Similarly, bullies were also found to have a positive correlation with anxiety. The findings of the meta-analysis also supported the hypothesis that victims of cyberbullying are more likely to suffer from anxiety compared to individuals who were not the target of such behavior [40].

Self-esteem refers to how individuals rate themselves and the degree to which they accept themselves [41]. According to Patchin and Hinduja [42] and O’Brien and Moules [43], victims of cyberbullying also experience a decline in their levels of self-esteem. The term “self-esteem,” which can be loosely translated as either a positive or negative attitude toward one’s own persona [44]. Another study [45] demonstrated that victims of cyberbullying report lower levels of self-esteem than non-victims of the same form of harassment online.

Internet addiction refers to an unhealthy preoccupation with using the Internet for a variety of activities, including those that are academic in nature, browsing the web, communicating via e-mail, chatting online, participating in social networking, and engaging in online entertainment [46]. Internet access, for both educational and recreational purposes, is now considered by most university students to be a crucial component of their day-to-day lives [47]. Nursing students use smartphones, computers, and personal digital assistants to download scientific research, and they use these devices in clinical settings, putting them at risk of Internet use issues [48].

Devi [49] reported that 20.8% of 231 nursing university students did not have Internet addiction, 64.9% had mild addiction, 14.3% had moderate addiction, and none had severe addiction. The degree of Internet addiction was statistically correlated significantly with a variety of factors, including having an active social media account, having an active email address, and daily surfing time [50]. Researchers found a correlation between the usage of the Internet and the occurrence of cyberbullying [51,52]. Addiction to the Internet and inappropriate use of the Internet are associated with cyberbullying [45].

There have been reports of an increase in the incidence of cyberbullying among adolescents residing in the Arab world; however, there are no exact numbers available, with the exception of a few studies that investigated the frequency of bullying in general. For instance, research conducted in Saudi Arabia indicated that the prevalence of cyberbullying ranged between 29.6% [53] and 31.5% [54].

According to our knowledge, this is the first study of its kind conducted among female nursing university students in Saudi Arabia to evaluate the relationships between various psychological factors and cyberbullying and cybervictimization. In Saudi Arabia and other Arab nations, there is scant literature on cyberbullying, cybervictimization and associated psychological issues. In addition, there is a paucity of research on the harmful consequences of cyberbullying and cybervictimization among university students, although this area of study is expanding. Therefore, the current study aims to estimate the prevalence of depression, low self-esteem, cybervictimization, anxiety, cyberbullying, and Internet addiction among Saudi female nursing university students and to identify the factors that predict cybervictimization and cyberbullying.

## 2. Materials and Methods

### 2.1. Design

In this particular research study, both a descriptive and a cross-sectional research approach were used to collect data. A cross-sectional study was chosen since it can yield results for a large number of participants in a short time. Participants in the study were female students studying nursing at a university in Saudi Arabia.

### 2.2. Sample and Study Population

In order to conduct a study with a significance level of 0.05, a medium effect size, and a statistical power of 0.95, it was found that there would need to be a minimum of 166 participants included in the investigation. This was the result of the calculation that was carried out in G Power 3.1.9.2, and it was the conclusion that was reached as a consequence of that computation. To account for a low response rate, the researcher can recruit more participants than needed. In most cases, an additional 20% to 30% of the estimated sample size is required [55]. Due to this, the researcher in the present study approximated that the sample size would consist of 199 Saudi female nursing university students. These students were chosen through the use of a convenient sampling method, and a non-response rate of 20% was taken into consideration. Participants needed to be actively enrolled in a nursing program and have the mental capacity to provide informed permission in order to take part in the study. Participants in the study who were enrolled in another medical area or participants whose responses to the socio-demographic questionnaire indicated they had a mental health diagnosis were excluded from further participation in the study.

### 2.3. Data Collection

The information was gathered using a variety of instruments, including the sociodemographic information, Rosenberg’s self-esteem scale (RSES), self-anxiety scale (SAS), the Chen Internet Addiction Scale (CIAS), WHO-Five Well-Being Index (WHO-5) and Cyber Victimization and Bullying Inventory.

#### 2.3.1. Sociodemographic Questionnaire

The researcher designed these questions as part of a questionnaire, and they were inspired by research that had already been done to identify the characteristics that were found to be important in similar studies.

#### 2.3.2. Rosenberg’s Self-Esteem Scale (RSES)

University students’ self-esteem was assessed using Rosenberg’s self-esteem scale (RSES). Each of the ten items in the RSES had four possible answers: “strongly agree,” “agree,” “disagree,” and “strongly disagree,” with corresponding scores of 1, 2, 3, and 4, respectively. Stronger scores indicated higher self-esteem, and the total RSES score varied from 10 to 40. The Cronbach’s coefficient for the RSES in this study was 0.928, indicating that the results were trustworthy [44].

#### 2.3.3. WHO-Five Well-Being Index (WHO-5)

This is a self-report questionnaire that asks respondents to score their state of well-being on a 6-point Likert scale (from 0 (At no time) to 5 (All time)) over the past two weeks. With a possible range from 0 to 25, higher WHO-5 scores indicate a greater sense of well-being. An inadequate health status and the need for additional testing for depression are indicated by a score of 12 or less [56]. The WHO-5 has been found to be reliable, valid, and able to identify people with depression in a wide range of settings across Asia, Australia, Africa, North America, Europe, and South America [57]. In this study, the WHO-5 has a Cronbach’s alpha of 0.856.

#### 2.3.4. The Chen Internet Addiction Scale (CIAS)

This is a four-point, 26-item self-reported scale. It measures five dimensions of Internet addiction, including compulsive use, withdrawal, tolerance, interpersonal interaction issues, and health and time management issues. The CIAS total score ranges from 26 to 104. The degree of Internet addiction increases as the CIAS score rises. The internal reliability of the scale and subscales in the original study varied between 0.79 and 0.93. Correlation analyses revealed a significantly positive connection between total-scale and subscale CIAS scores and the number of hours per week spent on Internet use [58]. The Cronbach’s coefficient for the CIAS in the current study was 0.826.

#### 2.3.5. Self-Anxiety Scale (SAS)

The self-anxiety scale (SAS) developed by Zung [59] is made up of 20 different Likert-scale items that are used to quantify the severity of various mental and physical symptoms. Each item was assessed by the participants on a 4-point scale, from 1 (never or seldom) to 4 (often or almost always). Items 5, 9, 13, 17, and 19, all of which represent a positive experience, had their ratings reversed. The raw score sum can be anywhere between 20 and 80. In order to create a “SAS Anxiety Index,” the raw score must be adjusted. To get the SAS Anxiety index, take the raw score and divide it by 80, then multiply the result by 100 [59]. Recent studies have suggested that the cutoff point be a raw score of 36 or an index score of 45 [60]. In this study, the SAS’s reliability coefficient was 0.896.

#### 2.3.6. Cybervictimization and Bullying Inventory

The inventory consists of 22 self-reported items and a Likert Scale with five points. Both victimization and bullying make use of the same tools and materials. The victimization section contains 22 items that begin with the phrase “I exposed” while the bullying section contains the same 22 items but begins with the phrase “I exhibit”. Analyses of both reliability and validity were carried out for the two different scales. The three variables that were identified in the inventory are known as cyber verbal bullying, obscuring identity, and cyber forgery [61]. The Cronbach’s coefficient for the Cybervictimization and Bullying Inventory in this study was 0.822.

### 2.4. Ethical Consideration

The Institutional Review Board (IRB) awarded the research project its approval. All of the female nursing students who participated in the study were made aware that they were under no obligation to answer any of the questions and might withdraw at any time if they so desired. Only overall results would be discussed to safeguard participants’ privacy.

### 2.5. Data Analysis

In order to analyze the data, version 20 of the Statistical Package for the Social Sciences (SPSS) was used, and these analyses were performed in three separate steps. First, descriptive statistics were used to depict the data on the students’ levels of depression, self-esteem, anxiety, cyberbullying, and cybervictimization. These measures were taken into account, these statistics included frequency and percentage analyses. Second, bivariate analyses using Chi (χ^2^) test were carried out in order to evaluate the likelihood of significant mean differences between demographics and cyberbullying and cybervictimization. These tests were carried out in order to investigate the plausibility of these disparities. Third, the Pearson correlation coefficient (r) was utilized in order to explore the potentially significant association between continuous variables such as depression, self-esteem, anxiety, Internet addiction, and cyberbullying, cybervictimization. This was done in order to determine whether or not there was a correlation between these factors. A Shapiro–Wilk test was performed to examine the normality of the data. The multivariable logistic regression model was then used to examine the independent variables, including demographic information, depressive symptoms, self-esteem, anxiety, and Internet addiction, that may predict the dependent variables, which are cyberbullying and cybervictimization., with results shown as the adjusted odds ratio (AOR) and 95% confidence intervals (CI). This study also evaluated the multicollinearity of the data and the Hosmer–Lemeshow test for the fit of the multivariable logistic regression model. The variance inflation factor (VIF) was employed to investigate the existence of multicollinearity. It was found that multicollinearity is not present when the mean value of VIF is less than 10. In addition, if the *p*-value of the Hosmer–Lemeshow test for goodness of fit for a multivariable logistic regression model was less than 0.05, this indicated that the model does not fit well. The VIF values in the current study were all less than 10 and the *p*-value of the Hosmer–Lemeshow test was 0.15.

## 3. Results

A total of 179 students (or 89.95%) participated in the survey by filling out questionnaires (179 out of 199). In total, the participants’ average age was 20.80 ± 1.62. Of the 101 participants who took part, 56.42% were between the ages of 18 and 20; 19.50% were married; 83.80% received support from their families; and 50.28% have more than three profiles on their preferred social media sites. Medically, 132 of the participants (82.11%) had no history of a chronic condition (See Table 1).

There were 19.55% of students who reported having low self-esteem, depression (30.17%), Internet addiction (49.16%), anxiety (34.64%), cyberbullying (20.67%), and cybervictimization (17.32%). Female nursing college students aged 18 to 20 suffered from low self-esteem (24.75%), depression (38.51%), Internet addiction (68.32%), anxiety (42.57%), cyberbullying (13.86%), and cybervictimization (10.89%).

The prevalence of depressive symptoms was higher among married participants (60.00%), as were the prevalence of Internet addiction (60.00%), anxiety (71.43%), low self-esteem (25.71%), and cyberbullying (32.43%) and cybervictimization (42.86%). The percentage of participants who reported having low self-esteem (6.00%), depression (19.33%), Internet addiction (43.33%), anxiety (26.67%), cyberbullying (8.00%), and cybervictimization (6.67%) when they have family support is lower in comparison to those who do not. In comparison to those without chronic disease, those with chronic disease had higher percentages of low self-esteem (80.00%), depression (73.33%), Internet addiction (86.67%), anxiety (93.33%), cyberbullying (40.00%), and cybervictimization (80.00%) (See Table 1 and Table 2).

The bivariate analysis in Table 2 shows that the number of students’ social media profiles is strongly related to their experiences of cyberbullying (χ^2^ = 33.657, *p* < 0.001) and cybervictimization (χ^2^ = 37.992, *p* < 0.001). In terms of other socio-demographic factors, there were no significant differences.

In Table 3, we see how self-esteem, depression, Internet addiction, and anxiety are all linked to cyberbullying and cybervictimization. Both depression (r = −0.26, *p* < 0.001) and low self-esteem (r = −0.42, *p* < 0.001) were found to have a significant negative connection with cyberbullying. Anxiety (r= 0.47, *p* < 0.001) and Internet addiction (r = 0.48, *p* < 0.001) are also positively correlated. Further, cybervictimization was found to be substantially linked negatively with self-esteem (r = −0.23, *p* = 0.001), and depression (r = −0.33, *p* = 0.001). Moreover, positive correlation with Internet addiction (r = 0.50, *p* = 0.001), and anxiety (r = 0.37, *p* = 0.001). keeping in mind that the WHO-5 is an inverse-scoring depression assessment, with lower scores indicating greater depression.

According to the findings of a binary logistic regression analysis, depression and the number of students’ social media profiles had no significant effect on either cyberbullying or cybervictimization (Table 4). The AOR indicated that the probability of being a cyberbully decreased with a 21.8% per unit increase in self-esteem, respectively, while the probability of being a cyberbully increased with a 2.8% per unit increase in Internet addiction. Additionally, the AOR of the logistic model indicates that students have a 4.7% higher probability of being a cyberbully for each point increase in their anxiety level.

A student’s likelihood of being a cybervictim decreases by 16% per unit as their self-esteem improves. Moreover, the likelihood of being a cybervictim increases with a 2.7% per unit increase in Internet addiction. The AOR of the logistic model also shows that students’ likelihood of being cybervictim increases by 4.2% for every unit increase in their anxiety level. The final logistic regression model was statistically significant (*p* < 0.001). The final model’s R2 and adjusted R2 were 0.354 and 0.343, respectively.

## 4. Discussion

In this study, the prevalence of depression, self-esteem, cybervictimization, anxiety, cyberbullying, and Internet addiction among Saudi female nursing university students was estimated, and the factors that predict cybervictimization and cyberbullying were identified. Studies that have been done among college students have shown that the roles played by students who are participating in cyberbullying—whether as bullies, victims, or both—may each have various consequences for physical and mental exacerbations [27].

### 4.1. The Prevalence of Variables under Study

According to the findings of the current study, the prevalence of cyberbullying was found to be 20.67%, which supported the findings of previous research addressing the prevalence rates of cyberbullying among college students in United States (21.9%) [62] and in Saudi Arabia (20.7%) [63]. However, contrary to the current study findings, other research carried out among college students in Turkey found the prevalence of cyberbullying was 57% [2] and in Canada, it was 5.1% [11]. According to the findings of a study conducted by Fleming and Jacobsen [62], the prevalence of cyberbullying among school students was 39% in Oman, 33.6% in Lebanon, and 32% in Morocco. It was reported that the prevalence in cyberbullying Jordan was 47% [64]. Moreover, a recent study that was carried out in Lebanon found that the incidence of cyberbullying was 53.4% [65].

It is important to recognize that cyberbullying can manifest differently in various countries due to factors such as cultural norms, access to and use of information and communication technologies, the availability of standardized instruments for measuring the phenomenon, the size and composition of the sample, and the emphasis placed on pre-university students, despite the widespread nature of the problem at the higher education. Few studies have utilized university populations in their research [1].

In the current study, 17.32% of participants reported being cybervictims. Similarly, researchers have claimed that the incidence of students victimized by electronic means during their higher education time could range from 5 to 40% [66,67,68]. Additionally, similar to other study findings, it was also discovered that 19% of the university students in the sample had reported being victims of cyberbullying [67]. Cybervictimization on college campuses has been the subject of far fewer studies. Our study indicated a significantly higher incidence in comparison to the prevalence rate in another study, which was 8.6% [69]. Our results also indicated a significantly lower rate than another study in Turkey conducted among college students, which found a prevalence of 55.3% [70]. However, 5.1% and 24.1% of university students are bullied and victimized online, respectively, according to research by Faucher, Jackson, and Cassidy [11]. In contrast to the current findings, greater prevalence rates were also reported in a sample of university students, revealing that 43.3% of the students claimed that they had been a victim of cyberbullying [66].

According to the findings of this study, 19.55% of female nursing students had symptoms of low self-esteem. Similarly, the prevalence rate of low self-esteem among college students in the southwest of Ethiopia was 18.1% [71]. Furthermore, our result is lower than the prevalence rate of low self-esteem among nursing students in Saudi Arabia (23.8%). The results of this study are significantly greater than those of earlier studies, which found that the prevalence rate of low self-esteem among nursing students was 2.5% [72,73].

The prevalence rate of Internet addiction in this study was 49.16%. Similarly, Alshehri, et al. [74] found that 45.3% of students in Saudi Arabia had moderate online addictions, while only 4% of students had severe addictions to the Internet. The current study found a lower prevalence rate of Internet addiction than previous research in Egypt, which found an overall rate of Internet addiction of 54.6% [75].The high incidence of Internet addiction among university students was related to the fact that these students have a lot of free time. They are always looking for methods to communicate over the Internet, and they utilize the Internet as a way to escape the sources of stress that come with university, such as exams and studying [76]. The use of different evaluation instruments and cutoffs, as well as disparities in cultural and socioeconomic contexts, may account for the discrepancies in reported Internet addiction prevalence among the aforementioned research, even among those done within the same nation.

According to the findings of this particular study the prevalence rate of depression is reported to be 30.17%, comparable in some way to the findings of a metanalysis which found that the prevalence of depression among female college students was 34% [77]. The current result is significantly lower than what was discovered in college students in Saudi Arabia, where the prevalence rate was determined to be 45% [78]. Our findings indicate a lower prevalence of depression compared to those of previous research, such as one that discovered the rate of depression to be 44% [79]. The current study indicated that the prevalence of anxiety was 34.64%. This finding is consistent with prior studies, which found a considerable prevalence of anxiety among students, ranging from 24.7% to 39.9% [80,81,82,83].

### 4.2. Predictors of Cyberbullying and Cybervictimization

Student anxiety, self-esteem, and Internet addiction were all found to be significant predictors of cyberbullying behavior and victimization in a logistic regression analysis. It was shown that students engaged in less cyberbullying when their anxiety levels were lower and that students were less likely to become cybervictims when their anxiety levels were lowered. Similarly, it was found that anxiety symptoms were a risk factor for both being a victim of cyberbullying and being a perpetrator of it on others. Some researchers hypothesized that anxiety was both a significant factor in determining one’s involvement in bullying and one of the outcomes of such involvement [84,85]. Compared to victims of traditional bullying, cyberbullying victims report more social problems, higher anxiety levels, and substance abuse [86].

Furthermore, Internet addiction was a risk factor for cyberbullying and for being a cybervictim. There are multiple possible explanations for this occurrence. First off, frequent Internet use was linked to a higher risk of cyberbullying [87]. Second, addiction to the Internet was described as “an impulse control disorder,” and it was discovered that it was linked to a wide variety of psychosocial issues, including cyberbullying [88]. Third, cybervictimization has been linked to Internet addiction, according to studies [89]. A study conducted in China demonstrated that there is a bidirectional relationship between cybervictimization and Internet use. Finally, it has been demonstrated that longer periods of time spent online (more than two hours per day) are associated with an increased risk of experiencing cyberbullying [90].

According to the findings of the current research, having poor self-esteem makes one more likely to engage in bullying behavior or be the victim of bullying behavior. In parallel, recent research studying the relationship between cyberbullying and self-esteem have found that victims of cyberbullying report decreased self-esteem [91]. Perhaps those with low self-esteem are more prone to be victimized, or perhaps the experience of victimization itself lowers one’s self-esteem [92]. On the other hand, it is discovered that adolescents with a high level of self-esteem are less likely to engage in bullying [93]. In contrast, there was no evidence of a causal link between being a victim of cyberbullying and having low self-esteem. Interestingly, research shows that the link between bullying perpetration and low self-esteem is far less stable. Evidence suggests that bullies have a wider range of self-esteem than nonbullies, with some being higher [94] and some being lower [95,96]. 

### 4.3. Implications

Insights gained from the present findings suggest taking specific and efficient steps to protect university students from participating in cyberbullying behaviors or becoming cybervictims. University students who are in need of psychological assistance in order to lower their levels of anxiety and maintain their mental health should have timely access to such assistance from qualified psychological counseling nurses or instructors, both offline and online. In addition, university students should be taught by their families and the administration of their colleges how to make responsible use of the Internet, and the students themselves should be encouraged to limit the amount of time they spend online each day. Professional psychological specialists also need to implement efficient intervention techniques for university students with an Internet addiction disorder in order to lessen the likelihood that these students may engage in cyberbullying or become cybervictims. The results of the current study should serve as a warning to policymakers, prompting them to take action to address cyberbullying, cybervictimization, and related issues.

### 4.4. Limitations

Though it was anticipated that this research would provide additional information regarding cybervictimization and cyberbullying among college students, readers must take the study’s limitations into account. First, this was a cross-sectional study, which means that the participants who took part were not followed up on over time. This problem might not occur if a long-term study was performed, but such a study would be expensive and would take a long time. Second, the fact that the sample was made up of only female participants made it harder to generalize the results. Additionally, the participants who took part in this research were undergraduates; therefore, the results can not be generalized to all university students. Finally, because the data were based on self-ratings, it is possible that there was some recall bias. Therefore, it is recommended to conduct systematic interviews.

## 5. Conclusions

In this cross-sectional study, we sought to explore the prevalence of anxiety, low self-esteem, depression, Internet addiction, cyberbullying, and cybervictimization among female nursing university students, as well as the factors that contribute to cyberbullying and cybervictimization. Students’ reported rates of low self-esteem, depression, Internet addiction, anxiety, cyberbullying, and cybervictimization were as follows: 19.55%; 30.17%; 49.16%; 34.64%; 20.67%; 17.32%. As a result, it can be stated that cyberbullying and cybervictimization were prevalent among nursing university students. The statistics presented above offered insights suggesting that focused and efficient efforts should be performed to prevent university students from engaging in cyberbullying or being victims of cyberbullying. It was discovered that student anxiety, self-esteem, and addiction to the Internet were all important predictors of cyberbullying behavior and victimization. Prospective studies should be carried out as a component of future research in order to provide further evidence of the impact that these risk factors have on instances of cyberbullying and cybervictimization. To further strengthen the validity of the findings, future research should aim for a larger sample size, multiple data collections on the same respondents, and the integration of self- and hetero-reports. The results can be used as a scientific benchmark for future intervention studies aimed at reducing instances of cyberbullying and cybervictimization among university students. The findings of this study can also be used as a rationale for future legislation meant to protect university students from both factors. Therefore, it is imperative that legislators and university officials take all of these variables into account when designing programs to combat cyberbullying and cybervictimization at the university level.

## Figures and Tables

**Table 1 ijerph-20-04293-t001:** Socio-demographic characteristics are described, together with the prevalence of low self-esteem, depression, Internet addiction and anxiety.

Study’s Variables	Total	(%)	Low Self– Esteem		Depression		Internet Addiction		Anxiety	
	N	%	N	%	N	%	n	%	n	%
Overall	145	100	35	19.55	54	30.17	88	49.16	62	34.64
Student’s Age/(years)										
18 to 20	101	56.42	25	24.75	39	38.61	69	68.32	43	42.57
≥21	78	43.58	10	12.82	15	19.23	19	23.36	19	24.36
Marital status										
Married	35	19.55	9	25.71	21	60.00	21	60.00	25	71.43
Single	144	88.45	26	18.06	33	22.92	67	46.53	37	25.69
Educational year										
1st to 2nd	102	56.98	23	22.50	39	38.24	68	66.67	42	41.17
3rd to 4th	77	43.02	12	15.58	15	19.48	20	25.97	20	25.97
Existence of chronic illness										
Yes	15	17.89	12	80.00	11	73.33	13	86.67	14	93.33
No	132	82.11	23	17.42	43	32.58	75	56.82	48	36.36
What is total number of profiles the you have across all of your favorite social media sites?										
1–3	89	49.72	8	8.99	13	14.61	34	38.20	29	32.58
≥3	90	50.28	27	30.00	41	45.56	54	60.00	33	36.67
Do you believe your family is supportive of you?										
Yes	150	83.80	9	6.00	29	19.33	65	43.33	40	26.67
No	29	16.20	26	89.66	25	86.21	23	79.31	22	75.86

**Table 2 ijerph-20-04293-t002:** The prevalence of cyberbullying and cybervictimization as well as the findings of bivariate analysis using the Chi (χ^2^) test.

Study’s Variables	Cyberbullying				Cybervictimization			
	n	%	χ^2^	*p*	N	%	χ^2^	*p*
Total	37	20.67			31	17.32		
Student’s Age/(years)			7.465	0.115			7.902	0.092
18 to 20	14	13.86			11	10.89		
≥21	23	29.49			20	25.64		
Marital status			3.554	0.084			6.973	0.013
Married	11	32.43			15	42.86		
Single	26	18.06			16	11.11		
Educational year			5.735	0.210			2.012	0.732
1st to 2nd	11	10.78			13	12.75		
3rd to 4th	26	33.77			18	23.38		
Existence of chronic illness			0.032	0.857			0.923	0.332
Yes	6	40.00			12	80.00		
No	31	23.48			19	14.39		
What is total number of profiles the you have across all of your favorite social media sites?			33.657	<0.001			37.992	<0.001
1–3	10	11.24			14	15.73		
≥3	27	30.00			17	18.89		
Do you believe your family is supportive of you?			0.173	0.676			2.937	0.568
Yes	12	8.00			10	6.67		
No	25	86.21			21	72.41		

**Table 3 ijerph-20-04293-t003:** Relationships between self-esteem, depression, Internet addiction, and anxiety with cyberbullying and cybervictimization.

Study’s Variables	Self-Esteem	Depression	Internet Addiction	Anxiety
Cyberbullying	−0.42 *	−0.26 *	0.48 *	0.47 *
Cybervictimization	−0.23 *	−0.33 *	0.50 *	0.37 *

* *p* < 0.001.

**Table 4 ijerph-20-04293-t004:** Predictors of cyberbullying and cybervictimization using a binary logistic regression analysis (n = 179).

Study’s Variables	*B*	SE	Adjusted odds Ratio (AOR)	*p*	95% Confidence Interval (CI)	
					Lower	Upper
Cyberbullying						
Self-Esteem	−0.212	0.031	0.782	0.002	0.830	0.950
Internet Addiction	0.028	0.012	1.028	0.003	1.012	1.049
Depression	−0.032	0.136	0.920	0.791	0.730	1.270
Total number of profiles across all social media sites	−0.126	0.068	0.042	0.592	0.870	1.120
Anxiety	0.046	0.010	1.047	<0.001	1.031	1.139
Cybervictimization						
Self-Esteem	−0.221	0.050	0.840	<0.001	0.810	0.920
Depression	−0.034	0.138	0.940	0.748	0.740	1.260
Internet Addiction	0.027	0.008	1.027	<0.001	1.010	1.042
Total number of profiles across all social media sites	−0.132	0.076	0.090	0.573	0.840	1.420
Anxiety	0.042	0.010	1.042	<0.001	1.030	1.066

## Data Availability

In response to a reasonable concern, the author will make the datasets used in this research accessible.

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
