# Peer review of "Anxiety, Depression, Self-Esteem, Internet Addiction and Predictors of Cyberbullying and Cybervictimization among Female Nursing University Students: A Cross Sectional Study"

_ijerph, 2023, doi:10.3390/ijerph20054293_

Round 1
Reviewer 1 Report
See attached file

Author Response
Thanks for your response and for giving me the chance to revise my manuscript.

Reviewer 2 Report
1. This study focuses on cyberbullying and cybervictimization, an important issue during the current world. This research has certain practical significance.
2. The study is reasonably designed, appropriately conducted, and has reliable results and conclusions, which can provide help and basis for the dampening of cyberbullying and cybervictimization.
3. There are still some improvements to be made in this study. For example, the references are not comprehensive and new enough. Therefore, more recent literature and empirical studies on cyberbullying and cybervictimization should be cited.
Author Response

(The authors gave the same response as above.)

Reviewer 3 Report
Dear Author(s),
I have read in detail your paper entitled "Anxiety, Depression, Self-Esteem, Internet Addiction and Predictors of Cyberbullying and Cybervictimization among Female Nursing University Students: A Cross Sectional Study".
The paper is written with a clear vocabulary (minor spell check required), follows a logical sequence and the English language is understandable and clear for the reader. Cyberbullying is a very common, ubiquitous and serious phenomenon among young people, so it is very important to conduct research in this area. The main aim of this study was to estimate the prevalence of depression, self-esteem, cybervictimization, anxiety, cyberbullying, and internet addiction among Saudi female nursing university students and to identify the factors that predict cybervictimization and cyberbullying.
The whole paper is very clear, logical and follows the structure of a scientific paper. However, to improve the quality of the paper a bit, I suggest the following improvements to the author(s):
Line 17 (abstract) + lines 265-266: When the percentages are listed first and then the constructs, it is a bit confusing for the reader because they have to keep scrolling back to see which order applies, i.e., which percentage "belongs" to which construct. I suggest writing the percentages next to the construct they belong to.
Introduction (lines 33-44): I suggest using current literature. This is a phenomenon that changes very quickly and very drastically, so references from 2009 or 2013 are not particularly relevant.
Lines 67-79: In this paragraph, I think it is important to mention, in addition to everything that has already been mentioned, the consequences of cyberbullying, which are very different from those of traditional bullying. And which can be much more difficult for the victim - the anonymity of the perpetrator, the lack of a safe place, the unlimited time of "abuse", difficulties in proving it, the material always remains "somewhere" in the online world. And this is also one of the arguments why it is necessary to study this. There is a lot of literature on this topic, and I believe that it will not be a problem for the authors to find it.
Lines 78-79: A reference is needed for the statement that significant prevalence rates were found in the academic context of universities.
Line 101: “such as”
Lines 160-161, 168-170, 206-207: a review of English needed. It's a bit unclear what this refers to. In general, I suggest checking the language throughout the paper.
MEASURE INSTRUMENTS: Other than the RSES, no other instrument used has a reliability coefficient for this sample shown in the paper. I suggest adding reliability coefficients everywhere where is applicable (specifically on WHO-5; CIAS)
In the "Results" section, before presenting the results of the logistic regression analysis, it must be stated whether the conditions for performing the logistic regression analysis are met, and the values must be given (at least I do not see that they were given in the paper). By this, I mean primarily multicollinearity and the Hosmer and Lemeshow test. If our preconditions are not fully met (or are on the "borderline"), we need to take this into account when interpreting the results but also mention it in the limitations of this study.
In addition, the presentation of the results of the regression analysis does not indicate the extent to which these predictors explain the criterion (what is the percentage of the variance of the criterion explained by these factors). And they are not apparent from the table either, and I think that is really valuable information for the reader. Of course, it's important to see the extent to which each of the included factors contributes but let us see the extent to which these predictors that were included (because the literature says they are "key" factors) contribute to explaining cyberbullying and cybervictimization.
In summary, the paper is very valuable, and with the suggested revisions, the paper will be of much higher quality and suitable for publication in this journal.
With respect,
Reviewer
Author Response

(The authors gave the same response as above.)
